# Effect of Translocation on Host Diet and Parasite Egg Burden: A Study of the European Bison (*Bison bonasus*)

**DOI:** 10.3390/biology12050680

**Published:** 2023-05-04

**Authors:** Christinna Herskind, Heidi Huus Petersen, Cino Pertoldi, Stine Karstenskov Østergaard, Marta Kołodziej-Sobocińska, Wojciech Sobociński, Małgorzata Tokarska, Trine Hammer Jensen

**Affiliations:** 1Department of Chemistry and Bioscience, Aalborg University, Fredrik Bajers Vej 7H, 9220 Aalborg, Denmark; 2Centre for Diagnostic, Technical University of Denmark, 2800 Kgs. Lyngby, Denmark; 3Mammal Research Institute, Polish Academy of Sciences, Stoczek 1, 17-230 Białowieża, Poland; 4Institute of Biology, University of Białystok, Ciołkowskiego 1J, 15-245 Białystok, Poland; 5Aalborg Zoo, Mølleparkvej 63, 9000 Aalborg, Denmark

**Keywords:** European bison, eDNA, parasites, diet, strongyles, Lille Vildmose, Białowieża Forest, EPG, Baermann, flotation, nanopore sequencing, rewilding, conservation

## Abstract

**Simple Summary:**

The aim of this study was to investigate the adaptability of ten recently introduced European bison (*Bison bonasus*) in Lille Vildmose. Populations from Bornholm, Denmark, and Białowieża Forest, Poland, were used as a reference. We investigated the adaptability of the European bison through analysis of their dietary diversity and parasitic load during twelve months after translocation. From March 2021 to February 2022, fecal samples were collected from the herds. In Lille Vildmose, a significant difference in egg per gram feces (EPG) was observed for June when compared to the months during autumn and winter (October–February). Metabarcoding of 63 bison dung samples collected during March–September in Lille Vildmose identified eight species of nematodes in the gastrointestinal tract of the European bison, with Haemonchus contortus being most frequently observed. Furthermore, 79 plant taxa were identified in the diet of the European bison. The broadest diet was observed in March suggesting that the bison quickly adapted to their new habitat. Additionally, the results suggested a seasonal shift in their diet, with this being most apparent from March to April.

**Abstract:**

For the purpose of nature management and species conservation, European bison (*Bison bonasus*) are being increasingly reintroduced into nature reserves across Europe. The aim of this study was to investigate European bison’s adaptability to new areas through the study of their parasite-EPG (eggs per gram feces) and dietary diversity during twelve months after translocation. We compared the parasite-EPG from introduced European bison in Lille Vildmose, Denmark, with the parasite-EPG from populations from Bornholm, Denmark, and Białowieża Forest, Poland. From March 2021 to February 2022, fecal samples were collected from three populations. Samples from Lille Vildmose were examined through flotation, sedimentation, the Baermann technique, and nanopore sequencing. Fecal samples from Bornholm and Białowieża were examined through flotation and sedimentation. Nanopore sequencing of DNA from 63 European bison’s fecal samples collected during March–September in Lille Vildmose identified 8 species of nematodes within the digestive tract of the European bison, with *Haemonchus contortus* being the most frequently observed. In Lille Vildmose, a significantly higher excretion of nematode-EPG was observed during the summer period than in the spring, autumn, and winter. In addition, monthly differences in the excretion of nematode eggs were found, with this being significantly higher in June than in the months during autumn and winter (October–February). Significant differences in the nematode-EPG were only found between the excretion of nematode eggs in Białowieża Forest when compared to that of Lille Vildmose, with significantly higher excretion in Lille Vildmose (October–November). The results indicate that the development rates for nematodes may be affected by changes in temperature, with increasing temperatures speeding up their development time. Independent of this study design, wildlife vets together with the gamekeepers managing the herd found it necessary to treat the herd with antiparasitics for practical and animal welfare reasons in relation to translocation. Furthermore, 79 plant taxa were identified in the diet of the European bison. The broadest diet was observed in March suggesting that the European bison quickly adapted to their new habitat. The results suggest a seasonal shift in their diet, with this being most apparent from March to April.

## 1. Introduction

The European bison (*Bison bonasus*) is the largest terrestrial mammal in Europe, with adult bulls occasionally weighing up to 1000 kg [1]. The population declined severely during the early twentieth century as a consequence of habitat fragmentation and overhunting. Eventually, the last lowland bison (*Bison bonasus bonasus*) in the Białowieża Forest died before the end of World War I. The last wild individual of Caucasian bison (*Bison bonasus caucasicus*) was killed in 1927 in Caucasia [2,3,4]. The contemporary population of European bison originates from only 12 founders who survived in captivity [3]. It is not homogeneous; eleven of these founders were lowland bison, while one bull was a Caucasian bison [3]. In 1952 the first captive-bred European bison were released in Białowieża Forest, Poland [3]. As a result of reintroduction programs and long-term conservation management, nowadays, the wild-living European population consists of around 6700 European bison scattered around 47 subpopulations [5,6]. In 2020, the assessors of the IUCN red list reclassified the European bison from vulnerable (VU) to near threatened (NT) [6]. Nonetheless, the European bison is still endangered, and revealing knowledge about its ecology and biology is essential for the species’ protection [7,8].

Large herbivores are increasingly reintroduced across Europe to promote biodiversity, restore ecosystem functioning, and contribute towards a rewilding process in both open landscapes and forest ecosystems [9,10]. The Danish megafauna consists mostly of deer species, which may result in an herb layer dominated by graminoids as deer usually avoid these. The European bison overlaps both intermediate feeders and grazers as a substantial part of their diet consists primarily of woody plants such as shrubs, shoots, and leaves [11,12]. Additionally, the diet of the European bison includes a diverse proportion of graminoids [10]. Therefore, reintroducing the European bison could result in the enrichment of the herb species layer compared to areas mainly grassed by deer species, as it has been observed that translocated animals tend to search for food that they are familiar with [8,9,10]. Furthermore, the dietary composition of the European bison varies depending on their habitat, including the plant biomass and the availability of plant species [13]. With this study, we sought to investigate the diet of ten recently introduced European bison in Lille Vildmose, Northern Denmark, in a fenced area with no public access, thereby gaining further knowledge of the adaptability of European bison to the diet in this area after translocation.

Health control and selective removal of sick animals is proposed as key components for the protection of the European bison [14]. The European bison is highly inbred and of dramatically poor genetic variation [15]. Consequently, the species may face problems of increased susceptibility to parasites, which may be additionally enhanced by releasing captive-bred individuals into the wild [16]. There is no evidence that highly pathogenic parasite species such as, for example, blood-sucking nematode *Ashworthius sidemi* or eye-infecting nematodes from the genus *Thelazia* cause the European bison population to decline, e.g., in Białowieża Forest [17,18]. However, a long-term study has shown that European bison are able to survive emerging and pathogenic parasite invasion and form a kind of balance after years of co-existence, suggesting that despite inbreeding, the European bison has the potential to cope with pathogens [17]. During the last century, more than 80 species of parasites have been recorded to infect European bison, including 4 trematode species and 43 nematode species [19]. *Dictyocaulus* sp. and *Fasciola hepatica* commonly infect European bison with a recorded prevalence of 34.2% (80/234) and 48.7% (114/234), respectively, for European bison examined in Białowieża Forest in 2008–2013 [3,19]. Additionally, the blood-sucking nematode *Ashworthius sidemi* is one of the most pathogenic parasites of European bison and other ruminants [17,20,21,22,23]. It was first found in European bison in Białowieża Forest in 2000 and still persists in the population [17].

In summary, the translocation of European bison individuals is associated with risks of introducing parasites to new areas, such as, i.e., *A. sidemi* [24]. However, most parasites hosted by European bison are also infectious to other bovids [25]. This allows the transmission of parasites between different animal species, which may pose a threat to the health of European bison and introduce parasites to the existing wildlife and potentially through wildlife to domestic animals. Furthermore, animals may experience an increased stress level when transferred to new areas, which can cause a decrease in immunity to parasite infections [24,26]. Based on this, we investigated the monthly changes in parasite-egg excretion in European bison translocated to Lille Vildmose (Denmark). Prior to translocation, the European bison were given antiparasitic treatment. The excretion of parasite eggs was compared with data obtained from well-established populations in Bornholm (Denmark) and Białowieża Forest (Poland).

## 2. Methods

### 2.1. Study Areas and Populations

In April 2021, 1- European bison were introduced to a 4000-ha large enclosure in Lille Vildmose (56.845° N, 10.214° E), Northern Jutland, Denmark. Besides European bison, the enclosure holds red deer (*Cervus elaphus*), roe deer (*Capreolus capreolus*), and wild boar (*Sus scrofa*). Three individuals of European bison were transferred from a smaller fenced area in Lille Vildmose where they were kept for acclimatization after life in Randers Rainforest, a zoo in Denmark. The other seven European bison were directly transferred from Holland (three from Lelystad Nature Park and four from Maarshorst Nature Reserve. Information on sex, birthdate, and origin is given in Table 1. 

In Lille Vildmose, the animals were supplementary fed with browser concentrate and hay (Granovit, Brogaarden, Denmark) from September 2021 to March 2022 to secure a sufficient body condition. On 21 April 2021, all ten animals were treated with antiparasitic treatment prior to transport from Holland and Randers to Lille Vildmose. The animals translocated from Randers Rainforest received Doramectin 10 mg/mL subcutaneously 0.2 mg/kg in 2019 prior to translocation to a smaller area of Lille Vildmose. Then they were treated with Eprinex 5 mg/mL pour on 0.5 mg/kg prior to moving from one part of Lille Vildmose to the final destination. Beyond that, the animals were treated with Curofen 50 mg/g perorally 7.5 mg/kg for 3 days from 22–25 October and from 24–27 November 2021. This was carried out as a strategy to prevent parasite infections.

The weather in Lille Vildmose is characterized by an oceanic climate. During 2021, the annual mean temperature was 8.2 °C, with July being the warmest month (18.1 °C) and February being the coldest (−0.5 °C) [DMI 2021].

The established European bison in the Bornholm population are located in a fenced area of 200 ha (55.134° N, 14.912° E) [9,27,28]. When sampled, the herd from Bornholm consisted of three cows, five bulls, and two calves. Besides European bison, the enclosure holds European fallow deer (*Dama dama)* and roe deer [28]. The weather in Bornholm is characterized by an oceanic climate, with the warmest annual mean temperature in 2021 being measured in July (16.9 °C) and the coldest being measured in February (1.0 °C) [DMI 2021]. The European bison population on Bornholm has previously been associated with a high parasitic load (2012) but is today considered as well established [28].

Furthermore, approximately 700 European bison from the Polish part of Białowieża Forest live on over 10.000 ha (23°31′–24°21′ E, 52°29′–52°37′ N). The enclosure in the Polish part of Białowieża Forest also holds wild boar, roe deer, and red deer [3]. The weather in Białowieża is characterized by both continental and Atlantic climates [15], with the warm and cold seasons being clearly marked.

### 2.2. Sample Collection

At Lille Vildmose, ten samples of fresh feces were collected monthly from March 2021 to February 2022. Additionally, samples were collected weekly in September and October 2021. At Bornholm, three samples were collected monthly from March to November 2021. In Białowieża Forest, ten fecal samples were collected in November, then in December 2021, and in February 2022. Individual fresh fecal samples were collected from the ground at each sampling stage from the herd in Lille Vildmose, Bornholm, and Białowieża Forest. It was impossible to collect fecal samples from the rectum or come close enough to the animals to observe which animal excreted which portion of the feces since the European bison in these three herds are wild living. Therefore, it is unknown from which animals each fecal sample originates.

Immediately after sampling, the feces were divided into two parts. One part was analyzed right away using the Baermann technique. The other part of the fecal samples was frozen at −18 °C within two hours following sampling and until analysis. In total, 237 samples were collected from the European bison. Of these, 180 were from Lille Vildmose, 27 were from Bornholm, and 30 were from Białowieża Forest. Unfortunately, it was not possible to sample equally at all three study locations for practical reasons.

### 2.3. Analysis

The fecal samples were thawed the same day as they were analyzed. All fecal samples were analyzed for gastrointestinal parasite eggs and oocysts by a modified McMaster technique described by Roepstorff and Nansen [29] with a sensitivity of 5 eggs/oocyst per g feces (EPG/OPG). Briefly, 56 mL of tap water was added to 4 g of feces. The mixture was homogenized with a glass spatula and strained through gauze. Approximately 10 mL was transferred into a centrifuge tube and centrifuged at 178× *g* for 10 min. The supernatant was discarded, and the pellet was resuspended in 3 mL of flotation fluid (saturated sodium chloride with 50 g glucose per 100 mL, specific gravity 1.27 g/mL). The solution was pipetted into two McMaster counting chambers (2 × 0.15 mL) and microscopically examined (100×). Trematode eggs were identified by a modified sedimentation technique described by Taylor et al. [30]. Briefly, 3 g of feces was homogenized with 50 mL of tap water and strained through gauze. Then, 15 mL of filtrate was transferred to a centrifuge tube and allowed to sediment for 10 min. Subsequently, the supernatant was discarded, and the sediment was re-suspended in 15 mL of tap water. Following an additional 10 min sedimentation period, the supernatant was once again discarded, and one drop of 5% methylene blue was added. The sediment was transferred to a McMaster chamber and examined microscopically at 100× magnification. Only fecal samples from Lille Vildmose were analyzed for lungworm larvae and were only analyzed in September and October by the Baermann technique [30]. Only samples from Lille Vildmose were analyzed as it is imperative to use a fresh fecal sample in a *Baermann* test.

### 2.4. eDNA Analysis

DNA-sequencing was conducted on 63 samples from Lille Vildmose to quantify the diet and parasites among the introduced herd (9 per month, Mar-Sep). Parasites from Lille Vildmose were quantified by DNA sequencing, as the morphological traits of the eggs are not sufficient for parasite species identification. For both diet and parasite analysis, DNA from approx. 100 mg of each fecal sample was extracted using the DNeasy Plant Mini kit (Qiagen GmbH, Hilden, Germany) and the protocol for purification of total dna from plant tissue following the manufacturer’s recommendations. DNA concentrations were measured with a Qubit^TM^ c 1x dsDNA HS Assay kit (Invitrogen, Waltham, MA, USA) on a Qubit 3 fluorometer (Invitogen, USA). The Trnl intron was selected for plant dietary analysis because of its highly conserved regions and its high taxonomic coverage. In addition, the trnl gene is favorable to use when analyzing highly degraded DNA, such as DNA from fecal samples. The trnl-gene was amplified using primer c: 5′-CGAAATCGGTAGACGCTACG-3′ and d: 5′-GGGGATAGAGGGACTTGAAC-3′ [31]. The second internal transcribed spacer (ITS-2) in the nematodes was amplified using the primer NC 1: 5′-ACGTCTGGTTCAGGGTTGTT-3′ and NC2: 5′-TTAGTTTCTTTTCCTCCGCT-3′ [32].

For each primer set, a PCR reaction was conducted in duplicates of 25 μL (PCRBIO 1x Ultra Mix (PCR BIOSYSTEMS), 400 nM of each primer, 10 ng of template DNA, and nuclease-free water. Amplification with the NC primer set was run under the following conditions: initial denaturation at 95 °C for 2 min, followed by 35 cycles of 15 s at 95 °C, 15 s at 52 °C, and 50 s at 72 °C and final elongation at 72 °C for 5 min. Amplification with the *trnL* -c/d primers was run under the following conditions: an initial denaturation at 95 °C for 2 min, followed by 35 cycles of 15 s at 95 °C, 15 s at 50 °C, and 50 s at 72 °C and final elongation at 72 °C for 5 min. A positive control and a negative control were included in both PCR reactions to ensure the quality of amplicon generation. The libraries were purified using CleanNGS (CleanNA, Waddinxveen, The Netherlands) with a sample:bead ratio of 1:0.8. The library concentration was measured using a Qubit^TM^ 1x dsDNA HS Assay Kit (Invitrogen, USA) on a Qubit 3 fluorometer (Invitrogen, USA), and the size of selected libraries was checked by Agilent 2200 TapeStation using ScreenTape D1000 (Agilent Technologies, Santa Clara, CA, USA).

Approximately 100 fmol of each PCR product from the same sample was pooled and subsequently barcoded, pooled equimolarly, DNA repaired and end-prepped, adapter-ligated, cleaned, and loaded onto a single MinION R9.4.1 using SQK-LSK110 with EXP-PBC096 following the manufacturer’s recommendations (Oxford Nanopore Technologies, Oxford, UK). The library was sequenced for 40 h.

### 2.5. Pre-Processing of Amplicon Sequencing Data

The raw reads were based-called using Guppy v.6.0.6 (https://community.nanoporetech.com, accessed on October 2021) and demultiplexed using Porechop v.0.2.3 (parameters–discard_middle–require_two_barcodes–barcode_threshold 85) [33]. The reads were then visualized using Nanoplot v1.24.0 and low-quality reads (<85% basecall accuracy, Q-score = 9) were filtered using Nanofilt v2.6.0 (de Coster et al., 2018) keeping reads between 200 and 900 bp. The reads were polished using minimap2 [34,35] and subsequently clustered into OTUs (97%) and denoised using VSEARCH v.2.13.4 (parameters–cluster_unoise–centroids) [36]. The OTUs were mapped using the blastn algorithm, BLAST+ v2.12.0 [37].

### 2.6. Statistical Analysis

Statistical analyses were performed using R-studio (R-studio v4.0.2.). The normality of our data was assessed using the Shapiro–Wilk normality test. As the data followed a non-normal distribution, non-parametric tests were used. The differences in the monthly excretion of parasite eggs/oocysts were assessed through the Mann–Whitney test. Due to the large number of tests performed on the same data, the *p*-values were corrected through Sequential Bonferroni.

The diet was analyzed through principal component analysis (PCA) and a heatmap, using the ampvis2 package for RStudio. Furthermore, the differences in diet diversity among the months were determined using both the Shannon and Simpson indexes.

## 3. Results

Neither lungworms nor liver flukes were identified in the fecal samples from the European bison from Lille Vildmose. Eggs of *F. hepatica* were identified in feces in the European bison from Bornholm and Białowieża Forest. The excretion of *F. hepatica* eggs in the European bison from Białowieża Forest peaked in November with a median excretion of 29 EPG. The *F. hepatica* egg excretion in the European bison from Bornholm peaked in July-November with median excretion of 29 EPG (Figure 1).

The nematode EPG showed monthly variations for the European bison from Lille Vildmose with the highest median EPG in June (200 EPG) followed by July (150 EPG) (March–February, Figure 2). No monthly variation in the median EPG was observed for the bison from Bornholm (50–100 EPG) and Białowieża Forest (0 EPG). The overall median EPG independently of the month was the highest in the European bison from Lille Vildmose (200 EPG). However, the EPG differed significantly between Lille Vildmose and Białowieża Forest (October–November, *p* = 0.00052). A low excretion of Eimeria oocysts was found in all three herds and no significant differences in OPG were observed between the three European bison herds (Figure 3). In Lille Vildmose, the highest non-significant excretion of Eimeria oocysts was found in November.

Nanopore sequencing of DNA identified seven species of gastrointestinal strongyles in European bison feces from Lille Vildmose: *H. contortus, Ostertagia ostertagi, Ostertagia leptospicularis, Trichostrongylus axei, Oesophagostomum venulosum, Cooperia oncophora,* and *Cooperia pectinata.* Parasite DNA from *H. contortus* was most frequently observed (67.5%). The occurrence of other nematode species did not exceed 11.5%.

Nanopore sequencing identified 79 plant taxa within the diet of the European bison from Lille Vildmose. In total, 26 molecular operational taxonomic units (mOTU) were classified to the species level, 41 were classified to the genus level, 7 were classified to the family level, 2 were classified to the order level, and 3 were classified to the class level. Dietary diversity was observed to be the highest in March (Shannon = 1.43, Simpson = 0.97), which was one month post release in Lille Vildmose. Birch species (*Betula*) (62.87%) occurred most often and were found with high profusion in all April-September samples (Figure 4). Other detected trees were European beech (*Fagus sylvatica*) (1.73%), alder (*Alnus* spp.) (1.21%), willow (*Salix*) (0.49%), and oak (*Quercus*) (0.36%). A high abundance of mosses occurred in the European bison’s diet, with mosses from the *Hypnales* order being the most abundant (3.71%). Principal component analysis (PCA) indicated a seasonal shift in their diet, with this being most apparent from March to April (Figure 5).

## 4. Discussion

The European bison from Lille Vildmose were not infected with *F. hepatica*, neither before nor after treatment with fenbendazole. *F. hepatica* eggs were excreted from European bison from Bornholm and Białowieża Forest. This could be due to the absence of its intermediate host, the mud snail (*Galba truncatula),* and the fact that the enclosure does not allow other wild ruminants to enter the area [38]. In addition, only a low excretion of *F. hepatica* eggs was observed in European bison in Białowieża Forest and Bornholm. However, in 2015, the European bison from Bornholm were highly infected with *F. hepatica* [28]. This could possibly be due to a lack of treatment prior to their translocation and the fact that the animals had been infected in Poland previous to their translocation because liver flukes are commonly found among European bison [19].

For the European bison from Lille Vildmose, significantly higher excretion of nematode eggs (EPG) was observed during the summer period, whereas the lowest excretion was observed during the winter. This observation is consistent with previous studies that have documented that egg excretion in ruminant feces is season dependent [39,40], correlating with temperature and humidity [39,40]. Parasite egg excretion in European bison may also be related to management practices, such as winter supplementary feeding, as supplementary feeding may cause an increase in the concentration of bison around feeding areas [22] though this was not observed in this study.

It has previously been concluded that strongyle-EPG varies with age, with young European bison displaying a higher excretion of eggs than adults [41]. Despite the young age structure in Lille Vildmose, the median egg count did not exceed 200 EPG. However, the individual EPG did. It is therefore unknown if the higher EPG in Lille Vildmose reflects the sampling of younger animals compared to Bornholm and Białowieża Forest. Even though the population density of European bison in Bornholm was much higher than in Lille Vildmose, no significant differences were observed in parasite EPG between these two populations.

European bison from Lille Vildmose were treated with both fenbendazole and doramectin/eprinomectin during the study period (Figure 2). Fenbendazole has previously been shown to lower parasite-eggs excretion up to six weeks post treatment [42]. *Haemonchus contortus* has previously been described to show resistance to fenbendazole treatment [43]. Our results support these findings, as *H. contortus* was the most frequently observed nematode in Lille Vildmose, and no significant decline in parasite egg numbers was observed after treatment. Furthermore, ivermectin has been proven to affect the parasite burden for more than two weeks after treatment [44]. In all of samples, a low excretion of parasite eggs was observed (EPG < 500) [45]. However, we cannot rule out the possibility that the applied treatments may have contributed to the low parasite burden observed during the months after translocation in Lille Vildmose. Larska and Krzysiak (2019) suggest that surveillance and health control are the key components for the protection of European bison. We did not observe an association between translocation and parasite egg burden; in addition, a significantly higher EPG was not observed until approximately four months after translocation. Neither do other studies of European bison find an association between translocation and parasite burden [46]. However, ineffective anti-parasitic treatment of European bison before their transport to new places poses a risk of introducing new species of parasites. *Ashworthius sidemi* was introduced by European bison translocated from Białowieża Forest to the Czech Republic [24]. Moreover, it should be taken into account that captive European bison released into the wild are more susceptible to parasitic invasions compared to animals spending their entire life in natural conditions [22].

All strongyles identified by DNA sequencing in Lille Vildmose have been previously recorded in European bison [19]. Based on metabarcoding of eDNA the dominant species of strongyles in Lille Vildmose was *H. contortus*. *Haemonchus contortus* can be highly pathogenic to ruminants, due to its blood-sucking habits [30]. Furthermore, metabarcoding identified 79 plant taxa within the diet of the European bison in Lille Vildmose. Other authors identified 71 (Bornholm) and 105 (Białowieża Forest) different plant taxa, respectively [10,12]. Our results indicate that European bison in Lille Vildmose forage on a broad variety of plants, comprising species from different habitats, which verifies their plasticity in plant selection behavior [9,10,12]. According to Wróblewski (1927) [47], who analyzed the last free-ranging herd of the 20th century, the European bison diet consisted of approximately 80% trees. Additionally, a study of European bison from Białowieża Forest revealed that woody species constituted 59.4% of their diet [12]. Our findings are consistent with cited research, as trees made up the majority of the diet also in Lille Vildmose (71.37%). Opposite to our findings, Kowalczyk et al. (2019) [12] found the highest dietary diversity during the summer period attributed to a peak in available plant species.

In this study, we observed a seasonal change in the diet content from March to September. The seasonal changes indicate a shift in plant availability or seasonal variations in food quality [12]. In Lille Vildmose, March was the month of the species’ richest diet. This could be explained by the fact that European bison forage more intensively during the spring, when they supplement the shortages of food in winter [3,48]. The size of the European bison’s mouth was observed to be limiting the degree of selectivity when foraging [49]. This could explain the large variety of plant taxa occurring with low abundance.

One of the purposes of European bison translocation to Lille Vildmose was to limit the presence of birch trees in the area [50]. High coverage with birch species could explain why it was the most frequently observed plant taxon in the European bison diet in Lille Vildmose. The presence of trees in their diet reflects a choice of plants with high nutritive value [10]. The other most abundant taxa were mosses; this was expected as the main habitat types in Lille Vildmose include moorland and woodland [50].

## 5. Conclusions

We did not find evidence for the hypothesis that the European bison parasite egg burden was affected by translocation. Our results indicate that development rates for nematodes may be affected by changes in temperature, with increasing temperatures speeding up their development time. There are seasonal factors that highly affect the diet of the European bison, with the translocation event having little or no impact on diet diversity. A broad variety of plants was observed in the European bison diet, indicating a high plasticity in European bison plant selection behavior.

## Figures and Tables

**Figure 1 biology-12-00680-f001:**
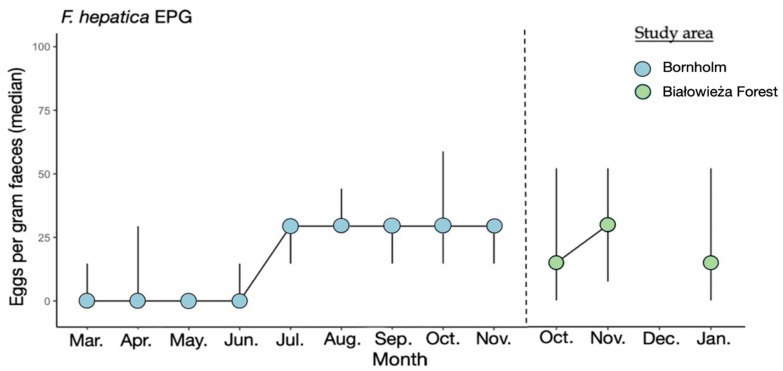
Median of *F. hepatica* eggs per gram feces (EPG) in European bison from Bornholm (n = 3) and Białowieża Forest (n = 10). No *F. hepatica* eggs were found in Lille Vildmose. The *x*-axis shows the months. The *y*-axis shows the median EPG for *F. hepatica.* The interquartile range (IQR) is given.

**Figure 2 biology-12-00680-f002:**
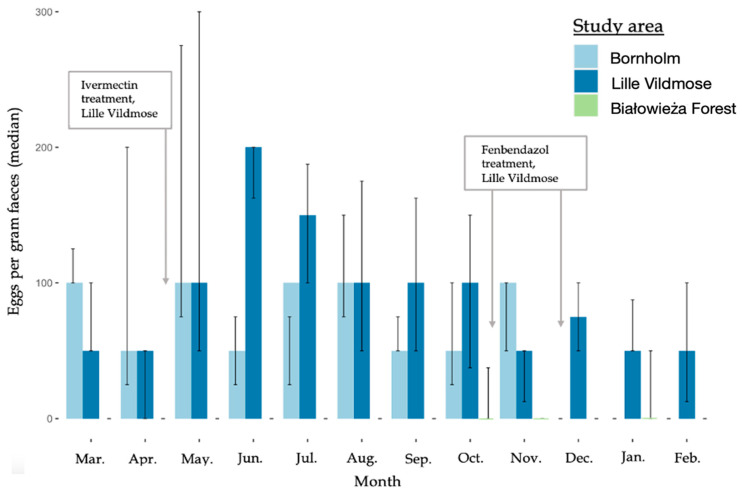
Median of nematode eggs per gram feces (EPG) in the European bison from Lille Vildmose (n = 10–40) and Bornholm (n = 3). The *x*-axis shows the months. The *y*-axis shows the median EPG for the nematodes. The interquartile range (IQR) is given.

**Figure 3 biology-12-00680-f003:**
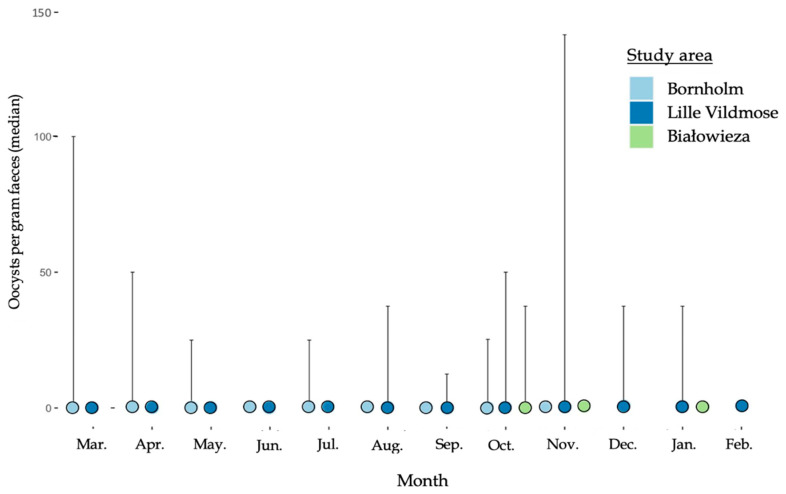
Median of *Eimeria* spp. oocysts per gram feces (OPG) in the European bison from Lille Vildmose (Nov–Aug, n = 10. Oct–Sep, n = 40), Bornholm (n = 3), and Białowieża Forest (n = 10). The *x*-axis shows the months. The *y*-axis shows the median OPG for the *Eimeria* spp. oocysts. The interquartile range (IQR) is given.

**Figure 4 biology-12-00680-f004:**
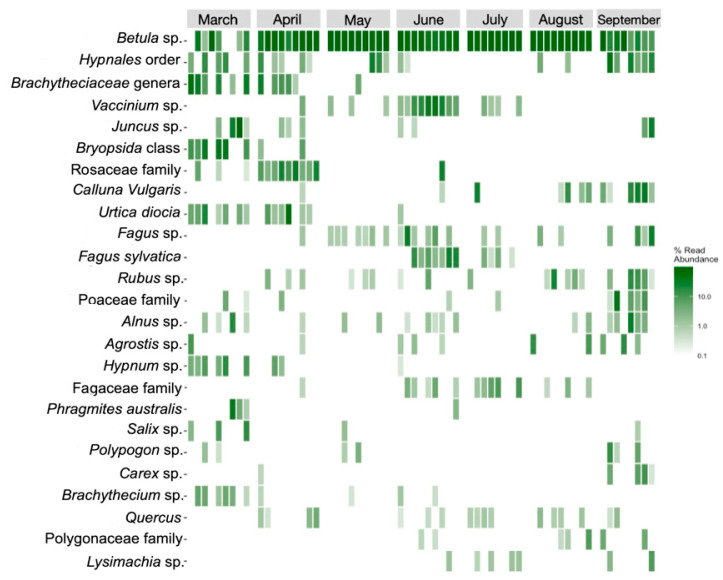
Heatmap showing the relative abundance of the plant taxa in the diet of European bison from Lille Vildmose. The 25 most abundant mOTUs (molecular operational taxonomic units) are shown. Each of the columns represents one sample. The color bar indicates the read abundance of each plant taxa.

**Figure 5 biology-12-00680-f005:**
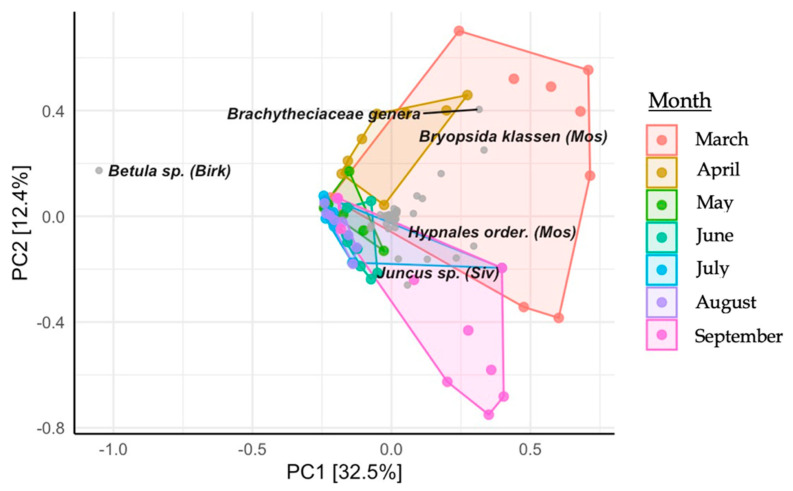
Principal component analysis (PCA) plot illustrating the monthly diet composition of the European bison from Lille Vildmose. The *x*-axis shows principal component 1 (PC1) and explains 32.5% of the variance. The *y*-axis shows principal component 2 (PC2) and explains 12.4% of the variance.

**Table 1 biology-12-00680-t001:** Gender, birthdate, and origin of the European bison from Lille Vildmose.

Gender.	Birthdate	Origin
Female	19 September 2011	Randers, Denmark
Female	9 May 2013	Holland
Female	1 June 2017	Holland
Male	3 September 2018	Randers, Denmark
Female	20 May 2019	Holland
Female	23 May 2019	Holland
Female	26 May 2019	Holland
Female	30 June 2019	Holland
Male	26 April 2020	Lille Vildmose, Denmark
Male	5 September 2020	Holland
Unknown	23 May 2021	Lille Vildmose, Denmark

## Data Availability

The authors confirm that the data supporting the findings of this study are available within the article.

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
