# Peer review of "Effect of Translocation on Host Diet and Parasite Egg Burden: A Study of the European Bison (Bison bonasus)"

_biology, 2023, doi:10.3390/biology12050680_

Round 1

Reviewer 1 Report

Hello,

I found the current study interesting and of current interest, since the  reintroduction of several animal species in natural habitats is more and more frequent. The importance of the European bison and its impact on natural habitats is important, since it helps to maintain the balance between several plant species. 

Abstract

I found the abstract, precise and it can stand alone with no problems.

Introduction

The introduction offers information about this animal species, and presents the current situation in Denmark. Some information regarding the current situation in Europe regarding this animals would be interesting to see in the introduction. Except for this remark, i find the introduction well written, and offers the necessary information for the reader.

Methods

Regarding study area and population, it would be nice to have more information about Bornholm and Bialowieza forest, as seen in case of Lille Vildmose.

Sample collection

Why the samples were not collected in the same way from all the three location?

Per total, the Methods section of the article, offers a good description of all the used methods. 

Results

This section presents the obtained results in a clean and understandable manner.

The discussion section is well written, and logical. Its easy to follow the line of thought.

Author Response

Thank you for your positive evaluation and good ideas which we will implement. Please find a point-point answer below

Some information regarding the current situation in Europe regarding this animals would be interesting to see in the introduction. Except for this remark, i find the introduction well written, and offers the necessary information for the reader.

We agree.

We added “As a result of reintroduction programs and long-term conservation management, nowadays wild-living European population consist of around 6700 European bison scattered around 47 subpopulations [5,6]. During 2020, the assessors of the IUCN red list reclassified the European bison from Vulnerable (VU) to Near Threatened (NT)[6].” In line 49 in the introduction.

Methods

Regarding study area and population, it would be nice to have more information about Bornholm and Bialowieza forest, as seen in case of Lille Vildmose.

Thank you for your comment. We have added: “Besides European bison, the enclosure holds: European fallow deer (Dama dama) and roe deer [27]. The weather on Bornholm is too characterized by an oceanic climate, with the warmest annual mean temperature in 2021 being measured in July (16.9°C) and the coldest in February (1.0°C) [DMI 2021]. The European bison population on Bornholm has previously been associated with a high parasitic load (2012) but are today considered well established [27]. Furthermore, approximately 700 European bison from the Polish part of the BiaÅ‚owieża Forest live on over 10.000-ha (23° 31´-24° 21´E, 52° 29´-52° 37´ N). The enclosure in the Polish part of the BiaÅ‚owieża Forest also holds: Wild boar, roe deer and red deer [3]. The weather in BiaÅ‚owieża is characterized by both continental- and atlantic climate [14], with the warm and cold seasons being clearly marked.” in line 118, after “calves.”: 

Sample collection

Why were the samples not collected in the same way from all the three location?

Unfortunately, we did not have direct access to Bornholm and Bialowieza so we were dependent on others to sample and it did not work out as we outline in our initial study plan. So, we had to do with what we got. We have now included at sentence page 3, line 134: “Unfortunately, it was not possible to sample equally at all three study locations for practical reasons”.

Reviewer 2 Report

General Comments:

This is a welcomed manuscript on the effect of translocation on host diet and parasite-EPG in a European bison population recently introduced in Denmark. However, a few omissions regarding the study design and results interpretation were identified. Furthermore, the literature data presented in the introduction is slightly disorganized. A few rephrases and some connections between the main ideas can be added in order to have a better understanding of the general picture. Several extra spaces, lack of commas, or other typos were identified (e.g. reacrion (line 170), fresh fecal sample in a Baermann (lines 154-155), salix (line 238)) and a careful re-checking of the manuscript is recommended.

Specific Comments:

Title: I would use “:” instead of “;”.

Abstract:

-          Line 26: Haemonchus contortus should be italicized.

-          Lines 26-29: I suggest combining the two sentences in one, as they are very similar.

-          Line 32: You also need to add a conclusion regarding the EPG in the abstract. What was the conclusion of your study?

Introduction:

-          Line 79: I suggest replacing “During the last century >80 species” with “During the last century, more than 80 species”.

-          I suggest adding information about the places of origin of the bison that were translocated to Lille Vildmose, including, if available, any previous studies regarding parasites of the original population.

-          Lines 94-95: I suggest rewording “translocated and given antiparasitic treatment”. Furthermore, this is important information that I would also mention in the abstract.

Methods:

-          Lines 109-112: I suggest adding more details like frequency (were those antiparasitic substances administered only once?), used doses, administration route, and commercial names of the substances.

-          Line 12-126: Is there any explanation for the frequencies you used for sample collection? It would have been better to use a similar approach for sample collection in all locations so you can have appropriate statistical results.

-          Lines 131-132: How about the samples tested with the Baerman technique? Please add all details regarding storage conditions in this section.

-          Lines 154-155: This was not mentioned in lines 131-132.

-          Lines 178-179: What does “amplicon generation” mean in this context?

Results

-          Lines 226-230: Please mention the full name of the genus at its first use. Moreover, I suggest mentioning what was the similarity between your DNA sequences and homologous sequences available in the GenBank® database.

-       Line 232: I recommend using the full name of “mOTU” instead of the abbreviation, as you used it just once in the whole manuscript.

-          Lines 235: As “birch” is the common name, you cannot add “spp.” after it. However, you can use ”species” instead.

Discussion

-          Lines 243-251: In the life cycle of Fasciola hepatica, the snail is not a vector, but an intermediate host. Moreover, you also need to consider, and, consequently, discuss in this section, that antiparasitic substances (especially fenbendazole) were used, and that the life cycle of this parasite lasts up to 20 weeks. Furthermore, at least 20 snail species were identified as intermediate hosts for Fasciola.

Supplementary files:

-          Figure 5: Is there any possibility you can italicize genus names?

Author Response

Thank you for your positive evaluation and good ideas which we will implement. Please find a point-point answer below

  • This is a welcomed manuscript on the effect of translocation on host diet and parasite-EPG in a European bison population recently introduced in Denmark. However, a few omissions regarding the study design and results interpretation were identified. Furthermore, the literature data presented in the introduction is slightly disorganized. A few rephrases and some connections between the main ideas can be added in order to have a better understanding of the general picture.

Yes, we have rephrased and made the necessary corrections.

  •  Several extra spaces, lack of commas, or other typos were identified (e.g. reacrion (line 170), fresh fecal sample in a Baermann (lines 154-155), salix (line 238)) and a careful re-checking of the manuscript is recommended.

We have now corrected the whole manuscript for extra spaces, commas and other typos.

Specific Comments:

  • Title: I would use “:” instead of “;”.

We agree. The semicolons have now been changed to ordinary colons.

Abstract:

  • Line 26: Haemonchus contortus should be italicized.
  • Lines 26-29: I suggest combining the two sentences in one, as they are very similar.
  • Line 32: You also need to add a conclusion regarding the EPG in the abstract. What was the conclusion of your study?

Thank you for the response on the abstract. We agree on your comments.

In line 26 Haemonchus contortus has now been italicized.

The sentences on line 26-29 have been changed to:

In Lille Vildmose, a significantly higher excretion of nematode-eggs was observed during the summer period than in spring, autumn and winter. Also, monthly differences in the excretion of nematode eggs were found, with June being significantly higher than the months during autumn and winter (Oct-Feb).”

 In line 32, the following conclusion has been added:

"The results indicate that development rates for nematodes may be affected by changes in temperature, with increasing temperatures speeding up the development time."

 Also, nematode-eggs has been changed to nematode-EPG (line 27)

Introduction:

  • Line 79: I suggest replacing “During the last century >80 species” with “During the last century, more than 80 species”

We agree. Line 79 has been corrected to “During the last century, more than 80 species”.

  • I suggest adding information about the places of origin of the bison that were translocated to Lille Vildmose, including, if available, any previous studies regarding parasites of the original population.

Yes, the places of origin is described page 3, line 103- 106. The status of parasites prior to translocation was low levels of strongylids eggs and since it was not us but external labs testing, we did not include these data (Trine H Jensen was involved in the translocations as a vet and Christinna actually doing these studies).

  • Lines 94-95: I suggest rewording “translocated and given antiparasitic treatment”. Furthermore, this is important information that I would also mention in the abstract.

 Yes, we reworded to: “Based on this, we investigated the monthly changes in parasite-egg excretion in European bison translocated to Lille Vildmose (Denmark). Prior to translocation the European bison was given antiparasitic treatment.”

In addition we added following in line 21 after “populations”.“Independent of this study design wildlife vets together with the game keepers managing the herd found it necessary to treat with antiparasitic for practical and animal welfare reasons in relation to translocation.

Methods:

  • Lines 109-112: I suggest adding more details like frequency (were those antiparasitic substances administered only once?), used doses, administration route, and commercial names of the substances.

Yes, we include it. The animals translocated from Randers Rainforest got Doramectin 10 mg/ml subcutaneously 0,2 mg/kg in 2019 prior to translocation to quarantine area of Lille Vildmose. Then they were treated with Eprinex 5 mg/ml pour on 0,5 mg/kg prior to moving from one part of Lille Vildmose to the final destination. Unfortunately, we did not pay attention to note what exactly the Dutch animals were treated with prior to translocation. “

We replace “fenbendazole” line 110 with “Curofen 50 mg/g perorally 7,5 mg/kg for 3 days.

  • Line 12-126: Is there any explanation for the frequencies you used for sample collection? It would have been better to use a similar approach for sample collection in all locations so you can have appropriate statistical results.

 Yes, we agree. Unfortunately, we did not have direct access to Bornholm and Bialowieza so we were dependent on others to sample and it did not work out as we outline in our initial study plan. So, we had to do with what we got. We have now included at sentence page 3, line 134: “Unfortunately, it was not possible to sample equally at all three study locations for practical reasons”.

  • Lines 131-132: How about the samples tested with the Baerman technique? Please add all details regarding storage conditions in this section.

We have now added the following in line 131 after “Originates”:

Immediately after sampling faeces was divided into two parts. One part was analyzed right away using the Baermann technique. The other part of the faecal samples were frozen at -18 °C within two hours following sampling and until analysis.”

  • Lines 178-179: What does “amplicon generation” mean in this context?

Generating DNA fragments through PCR, also called PCR products. By creating amplicons through PCR, we increase the number of a certain DNA-regions in our sample.

Results

  • Lines 226-230: Please mention the full name of the genus at its first use. Moreover, I suggest mentioning what was the similarity between your DNA sequences and homologous sequences available in the GenBank®

Line 226-230 was replaced with: Nanoporesequencing of DNA identified seven species of gastrointestinal strongyles in European bison faeces from Lille Vildmose: H. contortus, Ostertagia ostertagi, Ostertagia leptospicularis, Trichostrongylusaxei, Oesophagostomum venulosum, Cooperia oncophora and Cooperia pectinata.

  • Line 232: I recommend using the full name of “mOTU” instead of the abbreviation, as you used it just once in the whole manuscript.

We replace “mOTU” in line 232 with “molecular operational taxonomic units (mOTU)”

  • Lines 235: As “birch” is the common name, you cannot add “spp.” after it. However, you can use ”species” instead.

We agree. “birch spp” Has been changed to “birch species”

Discussion

  • Lines 243-251: In the life cycle of Fasciola hepatica, the snail is not a vector, but an intermediate host. Moreover, you also need to consider, and, consequently, discuss in this section, that antiparasitic substances (especially fenbendazole) were used, and that the life cycle of this parasite lasts up to 20 weeks. Furthermore, at least 20 snail species were identified as intermediate hosts for Fasciola.

Yes, it is a mistake. We correct to “intermediate host”. In Denmark fenbendazole is not labelled for treatment of Fasciola. But we do acknowledge it has been used before and in combination with other bendazoles. So, it is a good point. We will include in line 243 after F.hepatica “neither before or after treatment with fenbendazole”. “While” is deleted. We already wrote in line 249 that the treatment prior to translocation might have caused the negative results.:

Reviewer 3 Report

This study investigated the monthly changes in the parasites egg burden as well as diatary diversity in European bison which were translocated. Tranditional parasite examination and nanoporesequencing of DNA were used in this study. The writing english is very fluent in this manuscript. However, the study design and analysis need to be improved in this study. for example, the samples handling in method and analysis was too simple, which may not be qualified to submit in this journal.

1.      Would be possible that the host diet was correlated with parasite burden? more correlation study will be better to explain this topic.

2.      What is the infection rate of parasites in the investigated European bison?

3.      Line 21: Please clarify how many samples were collected in this study.

4.      Line 109:  Had all the animals been checked parasite burden before and after drug treatment?

5.      Line 131: All the samples were stored at -18 °C? Some parasite eggs/oocysts were damaged during the frozen procedure, for example, Eimeria oocyst. Therefore, the treatment of samples and analysis of EPG/OPG were not appropriate in this study. Maybe that was the reason why there were very low number of Eimeria oocysts in the results.

6.      Line 231:  How many bp of each OTU? Some of the OTUs were classified to species or genus. How to classified them?

7.      Line 234-240: Only the percentage of each plant taxa was not enough. Is it possible to compare these data statistically? As well as parasite burden.

Author Response

This study investigated the monthly changes in the parasites egg burden as well as diatary diversity in European bison which were translocated. Tranditional parasite examination and nanoporesequencing of DNA were used in this study. The writing english is very fluent in this manuscript. However, the study design and analysis need to be improved in this study. for example, the samples handling in method and analysis was too simple, which may not be qualified to submit in this journal.

  1. Would be possible that the host diet was correlated with parasite burden? more correlation study will be better to explain this topic.

Unfortunately, we did not observe any significant correlations (p > 0.05) between the parasitic load and the diet.

  1. What is the infection rate of parasites in the investigated European bison?

Unfortunately, we cannot measure it because we collected the feces from a semi wild herd so we do not know which individuals defecated the samples and if several samples could come from the same individuals. Therefore, we choose to compare the results within different herds.

  1. Line 21: Please clarify how many samples were collected in this study.

We updated the section “Sample collection”.

The following is described in line 131: “In total, 237 samples were collected from the European bison. Of these 180 were from Lille Vildmose, 27 from Bornholm and 30 from BiaÅ‚owieża Forest. Unfortunately, it was not possible to sample equally at all three study locations for practical reasons.”

  1. Line 109: Had all the animals been checked parasite burden before and after drug treatment?

Yes, we include it. The animals translocated from Randers Rainforest got Doramectin 10 mg/ml subcutaneously 0,2 mg/kg in 2019 prior to translocation to quarantine area of Lille Vildmose. Then they were treated with Eprinex 5 mg/ml pour on 0,5 mg/kg prior to moving from one part of Lille Vildmose to the final destination. Unfortunately, we did not pay attention to note what exactly the Dutch animals were treated with prior to translocation. “

We replace “fenbendazole” line 110 with “Curofen 50 mg/g perorally 7,5 mg/kg for 3 days.

  1. Line 131: All the samples were stored at -18 °C? Some parasite eggs/oocysts were damaged during the frozen procedure, for example, Eimeria oocyst. Therefore, the treatment of samples and analysis of EPG/OPG were not appropriate in this study. Maybe that was the reason why there were very low number of Eimeria oocysts in the results.

Yes, we are aware that we might lose some oocyst to freezing. However, this is a standardized way of testing Danish game approved by the Technical University of Denmark who was responsible for the national analysis during these studies. We had unfortunately only the possibility to freeze it before analysis regarding the samples from Bornholm and Poland thus we decided to do it to all samples to have it equal.

  1. Line 231: How many bp of each OTU? Some of the OTUs were classified to species or genus. How to classify them?

The amplicon length is given in the method-section.

The OUT was classified using the refseq (nt) database via blastN

  1. Line 234-240: Only the percentage of each plant taxa was not enough. Is it possible to compare these data statistically? As well as parasite burden.

Due to the low frequencies of some categories it was not possible to make statistical test based on the frequencies

Reviewer 4 Report

Please find the some comments on the manuscript:

1. Did the authors test the tap water before sample preparation for any contaminations?

2. It will be good to state why specifically trnl-gene was selected for diet analysis.

3. In Line 170, the sentence is not clear, maybe typing error.

4. Did the authors observe any correlation between diet and the burden of parasitic load in the animals?

Author Response

 Reply to Reviewer 3

Thank you for your positive evaluation and good questions. We have implemented the answers.

  • Did the authors test the tap water before sample preparation for any contaminations?

No, we did not but the tap water is drinking water for human consumption, so we do not expect it to be necessary.

  • It will be good to state why specifically trnl-gene was selected for diet analysis.

There is many advantages in using the whole trnl-intron and its P6-loop in diet analysis. The P6-loop has a short length, it is extremely variable, and it can be found in several copies per cell. For that reason, the P6-loop is favorable to use when amplifying highly degraded fragments of DNA, such as DNA from fecal samples. By the use of the primers trnl-g and trnl-h who binds to highly conservated regions of the trnl-gene, it is possible to amplify the P6-loop. Additionally, the primer pair trnl-c and -d may be used to amplify the whole trnl-gene.

We added “The Trnl intron was selected for plant dietary analysis because of its highly conserved regions and its high taxonomic coverage. Also, the trnl-gene is favorable to use when analyzing highly degraded DNA, such as DNA from fecal samples” in line 163, after “(Invitogen, USA).”

  1. In Line 170, the sentence is not clear, maybe typing error.

 Yes, we have improved it and corrected the spelling error “reactions”

  1. Did the authors observe any correlation between diet and the burden of parasitic load in the animals?

Unfortunately, we did not observe any significant correlations (p > 0.05) between the parasitic load and the diet.

Reviewer 5 Report

This is a very interesting study, nicely presented. I do not know if any calves were born in the study period? This might potentially have an influence on EpG. Also the individual variation between animals might explain some of the effects observed. I do agree with the author's conclusions, however, and do not have any concerns about this study. I think this is a very interesting read!

Author Response

Thank you for your positive evaluation and good questions. We have implemented the answers.

I do not know if any calves were born in the study period? This might potentially have an influence on EpG.

Yes, one calf was born in the study period, so we can not make any conclusions about the calving effect.

Also the individual variation between animals might explain some of the effects observed. Yes, this is absolutely true, however, the herd was not managed on individual basis and as stated page 3 line 127-132 Feces were collected from the ground and it was unfortunately not possible to identify from which animal the feces came so we could not study individual variation

Round 2

Reviewer 3 Report

no more comments

Author Response

  1. Would be possible that the host diet was correlated with parasite burden? more correlation study will be better to explain this topic.

Unfortunately, we did not observe any significant correlations (p > 0.05) between the parasitic load and the diet.

  1. What is the infection rate of parasites in the investigated European bison?

Unfortunately, we cannot measure it because we collected the feces from a semi wild herd so we do not know which individuals defecated the samples and if several samples could come from the same individuals. Therefore, we choose to compare the results within different herds.

  1. Line 21: Please clarify how many samples were collected in this study.

We updated the section “Sample collection”.

The following is described in line 131: “In total, 237 samples were collected from the European bison. Of these 180 were from Lille Vildmose, 27 from Bornholm and 30 from BiaÅ‚owieża Forest. Unfortunately, it was not possible to sample equally at all three study locations for practical reasons.”

  1. Line 109: Had all the animals been checked parasite burden before and after drug treatment?

Yes, we include it. The animals translocated from Randers Rainforest got Doramectin 10 mg/ml subcutaneously 0,2 mg/kg in 2019 prior to translocation to quarantine area of Lille Vildmose. Then they were treated with Eprinex 5 mg/ml pour on 0,5 mg/kg prior to moving from one part of Lille Vildmose to the final destination. Unfortunately, we did not pay attention to note what exactly the Dutch animals were treated with prior to translocation. “

We replace “fenbendazole” line 110 with “Curofen 50 mg/g perorally 7,5 mg/kg for 3 days.

  1. Line 131: All the samples were stored at -18 °C? Some parasite eggs/oocysts were damaged during the frozen procedure, for example, Eimeria oocyst. Therefore, the treatment of samples and analysis of EPG/OPG were not appropriate in this study. Maybe that was the reason why there were very low number of Eimeria oocysts in the results.

Yes, we are aware that we might lose some oocyst to freezing. However, this is a standardized way of testing Danish game approved by the Technical University of Denmark who was responsible for the national analysis during these studies. We had unfortunately only the possibility to freeze it before analysis regarding the samples from Bornholm and Poland thus we decided to do it to all samples to have it equal.

  1. Line 231: How many bp of each OTU? Some of the OTUs were classified to species or genus. How to classify them?

The amplicon length is given in the method-section.

The OUT was classified using the refseq (nt) database via blastN

  1. Line 234-240: Only the percentage of each plant taxa was not enough. Is it possible to compare these data statistically? As well as parasite burden.

Due to the low frequencies of some categories it was not possible to make statistical test based on the frequencies

Jeg kender ikke svaret på dette.